# Using somatic variant richness to mine signals from rare variants in the cancer genome

Saptarshi Chakraborty [1], Arshi Arora[1], Colin B. Begg [1]* & Ronglai Shen[1]*

To date, the vast preponderance of somatic variants observed in the cancer genome have been rare variants, and it is common in practice to encounter in a new tumor variants that have not been observed previously. Here we focus on probability estimation for encountering such hitherto unseen variants. We draw upon statistical methodology that has been developed in other fields of study, notably in species estimation in ecology, and word frequency estimation in computational linguistics. Analysis of whole-exome and targeted panel sequencing data sets reveal substantial variability in variant "richness" between genes that could be harnessed for clinically relevant problems. We quantify the variant-tissue association and show a strong gene-specific, lineage-dependent pattern of encountering new variants. This variability is largely determined by the proportion of observed variants that are rare. Our findings suggest that variants that occur at very low frequencies can harbor important signals that are clinically consequential.

---

[1] Department of Epidemiology and Biostatistics, Memorial Sloan Kettering Cancer Center, 485 Lexington Ave., 2nd floor, New York, NY 10017, USA. *email: beggc@mskcc.org; shenr@mskcc.org

dentifying somatic mutations in cancer genes is critical for precision oncology. The Cancer Genome Atlas (TCGA) project has cataloged somatic variants in over 10,000 tumor samples across 33 cancer types using whole-exome sequencing strategies[1]. With the advances in precision medicine programs driven by genomic testing, there has been a rapid growth in the availability of clinical sequencing data for cancer patients, including the MSK-IMPACT assay which has tested over 10,000 metastatic tumor samples[2]. These emerging datasets cataloging the distribution of somatic mutations across genes in tumor samples have demonstrated unequivocally a long-tailed distribution in which a relatively small number of variants appear in tumors frequently, but the vast majority of variants occur extremely infrequently.

Somatic variant analysis has been largely focused on known cancer-associated genes with frequent occurrences. However, this only utilizes <1% of the information from sequencing data sets and the vast trove of rare variants are "hidden" and unexplored. It is particularly striking that over 90% of variants in TCGA are singletons, i.e. variants observed only once in the >10,000 samples. In this study, these extremely rare variants are precisely what we focus on. We propose methods for mining the potential hidden iceberg of information in these variants and in the larger number of variants that have not yet been observed.

To address this issue quantitatively we focus on a novel problem of probability estimation for encountering new variants in a future tumor sample given the information from existing sequenced tumor cohorts. Analyzing data in this context is analogous to text mining in natural language processing applications. Consider, for example, the Google corpus which has more than 1 trillion words, with most words appearing infrequently. Statistical and machine learning techniques that utilize such training data for solving problems such as text prediction, machine translation, and other natural language processing problems, require appropriate handling of words or word sequences that are not present in the data. Accurate estimation of probabilities of encountering such hitherto unseen words often critically affect the accuracy of the final results. Drawing the analogy in cancer genomics, accurate estimation of the probabilities of encountering previously unseen variants could be valuable for improving the classification of the site of origin for cancers of unknown primary, the identification of clonal origin of metastasis, or "liquid biopsy" screening of circulating tumor DNA if these probabilities differ markedly by tissue of origin. In these settings, classical likelihood-based inference procedures for variant probability estimation is unsuitable because unseen variants in the training data are assigned zero probabilities. To overcome this challenge we draw upon ideas that have been studied over many years in a variety of scientific problems totally unrelated to tumor genomics, stemming from work to estimate the number of unknown or unseen species in an animal population[3], in related work to identify frequencies of individual words in linguistic studies[4], and more recently in estimating immune receptor diversity[5], complexity of genomic sequencing[6], and human genetic variations[7].

We apply a combination of these methods to the TCGA dataset encompassing whole-exome sequencing of ~10,000 tumors and validate the results on a clinical cohort of close to 10,000 tumors sequenced by a targeted cancer gene panel. Our analysis systematically maps probabilities of mutations and co-mutations across tissue types. The output provides a critical roadmap for developing novel diagnostic tools in identifying tumor origin from unknown primaries or in liquid biopsy settings.

## Results

**Data sources.** Our study uses two publicly available data sets: the somatic mutation data set from the TCGA 10,295 whole-exome sequencing study[1], and the Zehir et al[2]. MSK-IMPACT somatic mutation data set from targeted sequencing of 9593 tumor samples. MSK-IMPACT is a hybridization capture-based NGS clinical assay that is capable of detecting mutations in all exons and selected introns and promoter mutations in 410 (and, more recently, 468) cancer-associated genes. The composition of tumor types in each cohort and the sample sizes associated with each tumor type are shown in Supplementary Tables 1 and 2 for TCGA and MSK-IMPACT respectively.

**Methods overview.** We address the problem of estimating the probability of encountering a new non-synonymous single nucleotide variant (SNV) in a future tumor sample given the information from an existing sequenced tumor cohort. To accomplish this, we propose the use of the Good-Turing frequency estimator adapted for this specific problem, originally formulated by Turing and Good[8,9]. This approach recognizes that in observing events from a set of possible outcomes, if some of the outcomes are not observed then a proportion of the total probability needs to be reserved for these outcomes, a feature that is not present, for example, in maximum likelihood estimation. Good and Turing[8] devised an elegant strategy for estimating this missing probability. Briefly, the basic principle of the method is to construct a variant frequency vector, as measured through the pairs $(r, N_r)$ with $N_r$ denoting the number of variants appearing exactly r times. For example, $N_1$ is the number of variants appearing only once ($r = 1$) in the given cohort (i.e. frequency of singleton variants), $N_2$ is the number of variants appearing twice ($r = 2$) (i.e. frequency of doubleton variants), and so on. The Good-Turing estimator of the probability of occurrence in a randomly selected new tumor of a variant that has been observed r times in m previous tumors takes the following form:

$$q^{GT} = \frac{r+1}{m+1} \frac{S(N_{r+1})}{S(N_r)}$$

where $S$ denotes a smoothing function of $N_r$. We used a combination of different smoothing techniques in our analyses (see Supplementary Methods). The Good-Turing approach groups the variants with the same recurring frequency $r$, by assuming they occur with the same probability, and enables more effective estimation of the rare variant probabilities.

The preceding formula provides a straightforward solution for variants observed at least once previously ($r \geq 1$). However, use of this formula for estimating the probability for a variant that has not yet been observed requires knowledge of $N_0$, the total number of unseen non-synonymous SNV variants, an unknown quantity. We circumvent this problem in two ways. First, we make use of the fact that the probability of observing any one or more previously unseen variants in a new tumor can be approximated by the formula $1 - \exp\left\{-\frac{N_1}{m+1}\right\}$, a formula that does not involve the unknown $N_0$ Second, we recognize that the task of predicting the number of unseen variants is analogous to the species richness estimation problem in ecology, where the aim is to estimate the total number of unseen species present in a closed population. The most popular statistical model used for this task is the extrapolation approach first introduced in Fisher et al[3]. Here one first observes the incidences of variants ("species") in $m$ tumors, and then, based on the observed distribution of variants, one considers the problem of estimating/predicting the number of new variants ("species"), denoted $\Delta(t)$ that would be observed if $tm$ additional tumors outside the original sample were sequenced. Note that in our genomics setting $m$ represents the number of tumors profiled (sampling units) and $tm$ represents the number of tumors to be observed in the future sample. We have elected to use a smoothed version of the estimator proposed by Good and

Toulmin[4,10,11] for this purpose (defined in the Supplementary Methods). Both of these methods, our estimate of the probability that at least one new variant (in a gene) will be observed in a future tumor and our estimate of the number of new variants that will be observed in a specific number of future tumors, allow for direct empirical validation. For both we use the TCGA data to estimate these quantities and our validation dataset to evaluate their accuracy.

Since the number of variants in a tumor can by influenced strongly by hypermutation we stratified the tumors using a mutational signature analysis[12], and categorized each tumor into one of six categories: non-hypermutated, APOBEC (apolipoprotein B mRNA editing enzyme catalytic polypeptide-like), Smoking-associated, MMR (mismatch repair), UV (ultraviolet), and POLE (DNA Polymerase Epsilon, Catalytic Subunit). The five single base substitution (SBS) signatures were assigned according to the Sanger COSMIC mutational signature annotation[12]. (See Supplementary Methods for more details.) In the TCGA data set, the non-hypermutated group was defined as the set of tumors not in these five SBS signature categories, and with a total mutation burden <500 each[13]. A total of 140 TCGA tumors with mutation burden >500, but not in any of these six categories were excluded from our analyses. For the MSK-IMPACT data set, a hypermutation threshold of 38 was obtained by multiplying the TCGA threshold (500) with the ratio of median total mutation burden in MSK-IMPACT (4) to that in TCGA (54). The non-hypermutated tumor group in MSK-IMPACT dataset thus involved tumors not in one the aforementioned five SBS signature categories, and with total mutation burden <38. A total of 130 MSK-IMPACT tumors not in any of these six categories were excluded.

We use the Normalized Mutual Information (NMI) to quantify variant-tissue dependencies. NMI is a term that measures the extent to which the estimated probabilities of observing previously unseen mutations varies by tissue type. A value of zero indicates that the occurrence of the variant is lineage independent (equally likely to occur in any tissue type). By contrast, a large NMI value indicates that the variant occurs in a strongly lineage-dependent manner. For evaluating the strength of the association between estimated probabilities of observing previously unseen variants or the numbers of unseen variants and their observed frequencies in the validation dataset we used Lin's concordance correlation coefficient[14].

A detailed description of these methods (including some derivations) is included in Supplementary Methods. An R package containing datasets and software implementation of the methods used in this study has been released in the public domain (https://github.com/c7rishi/variantprobs).

**Predicting the number of unseen variants in a future sample**. The TCGA somatic mutation data set was derived from whole-exome sequencing of 10,295 tumor samples (across 33 cancer types), of which 10,275 have at least one non-synonymous SNV[1]. The tumor type composition is shown in Fig. 1a, further organized into broader categories of anatomic locations. A total of 1,788,153 unique somatic variants (here we focus on non-synonymous mutations) were detected, ~92% of them singletons (appearing only once in the cohort). In contrast four variants (*BRAF* V600E, *IDH1* 132H, *PIK3CA* E545K, and *PIK3CA* H1047R) appeared more than 200 times each (Fig. 1b). As discussed in Methods, the smoothed Good-Toulmin method can be used to estimate $\Delta(t)$, the expected number of new variants in a future sequencing cohort of size $\sim 10^4 t$ for various values of multiplying factor $t > 0$.

The distinctiveness of the mutational subgroups with respect to the numbers of new variants likely to be observed in a future sequencing cohort is displayed in Fig. 1c. The bar graph displays the total number of new variants expected throughout the exome in a single new tumor estimated using the Good-Toulmin formula. Due to the wide variations observed we have performed parallel analyses on each of these six tumor categories. We primarily focus on the results obtained from the non-hypermutated group, with results of relevant analyses for the other sub-groups presented in the supplement.

The MSK-IMPACT data set was derived from an FDA-approved targeted sequencing panel of 410 cancer-associated genes applied to prospectively sequenced tumors from close to 10,000 cancer patients[2]. The tumor type composition is shown in Fig. 1d depicting a relatively comparable composition with the TCGA cohort (Fig. 1a), with a few exceptions including pancreatic cancer ($m = 463$ MSK-IMPACT vs. $m = 176$ in TCGA), lung adenocarcinoma ($m = 1194$ MSK-IMPACT vs. $m = 568$ TCGA), and colorectal cancer ($m = 969$ MSK-IMPACT vs. $m = 559$ TCGA) (Supplementary Tables 1 and 2). The larger numbers of the three cancer types in the MSK-IMPACT cohort are consistent with a higher number of *KRAS* G12D, G12V, and G12C variants observed in the MSK-IMPACT cohort (Fig. 1e). A total of 46,806 unique somatic variants (non-synonymous) are detected, 90% of which are singletons. Figure 1f displays the classification of these tumors into the subgroups defined by mutational signatures, showing a broadly similar breakdown to the TCGA data. Of note, the horizontal axis in this figure displays the expected number of previously unseen variants in a new tumor in the 410 genes in the MSK-IMPACT panel versus the total in the exome displayed in Fig. 1c. We note that mutation burden among the 410 genes by tumor type is largely similar in the two cohorts (Supplementary Fig. 1).

Focusing on the non-hypermutated cases and using TCGA as the training data set and MSK-IMPACT as a "prospective" validation data set, we show that encountering new variants in a prospective cohort (e.g., MSK-IMPACT) that have not been observed in the training cohort (TCGA) is an extremely common event. Figure 2a shows the average number of variants detected in a tumor by tissue sites (blue) and the average number of new variants in a tumor (orange) in the MSK-IMPACT cohort, i.e. variants observed in MSK-IMPACT but not in TCGA, alongside the number of new variants predicted using the TCGA training data (maroon) for the most frequent cancer types. It is striking that for most tumor types, over 60% of the variants detected in a tumor are new ones that have not been observed in the TCGA cohort. A raincloud plot[15] in Fig. 2b displays the distribution of the new to total variant ratio in the "prospective" cohort at the individual tumor sample level. This suggests that future efforts of sequencing a larger number of tumors will inevitably lead to the identification of new variants. In the next section, we further show that many new variants in specific genes emerge in a tissue-dependent manner.

**Probability estimation reveals a specific tissue-type pattern**. In this section, we focus on the 6696 non-hypermutated tumors in TCGA and turn our attention to the estimation of the probabilities of individual mutations occurring in a gene, with major emphasis on rare and hitherto unobserved mutations. Supplementary Figure 2 displays the frequency vectors for three cancer genes with contrasting patterns: *PTEN*, *FAT1*, and *KRAS*. We see for example that *KRAS* has 24 singletons, 3 doubletons, and a few hot-spot variants that occur at very high frequencies including the G12D (appearing $r = 142$ times), G12V ($r = 120$ times), and G12C

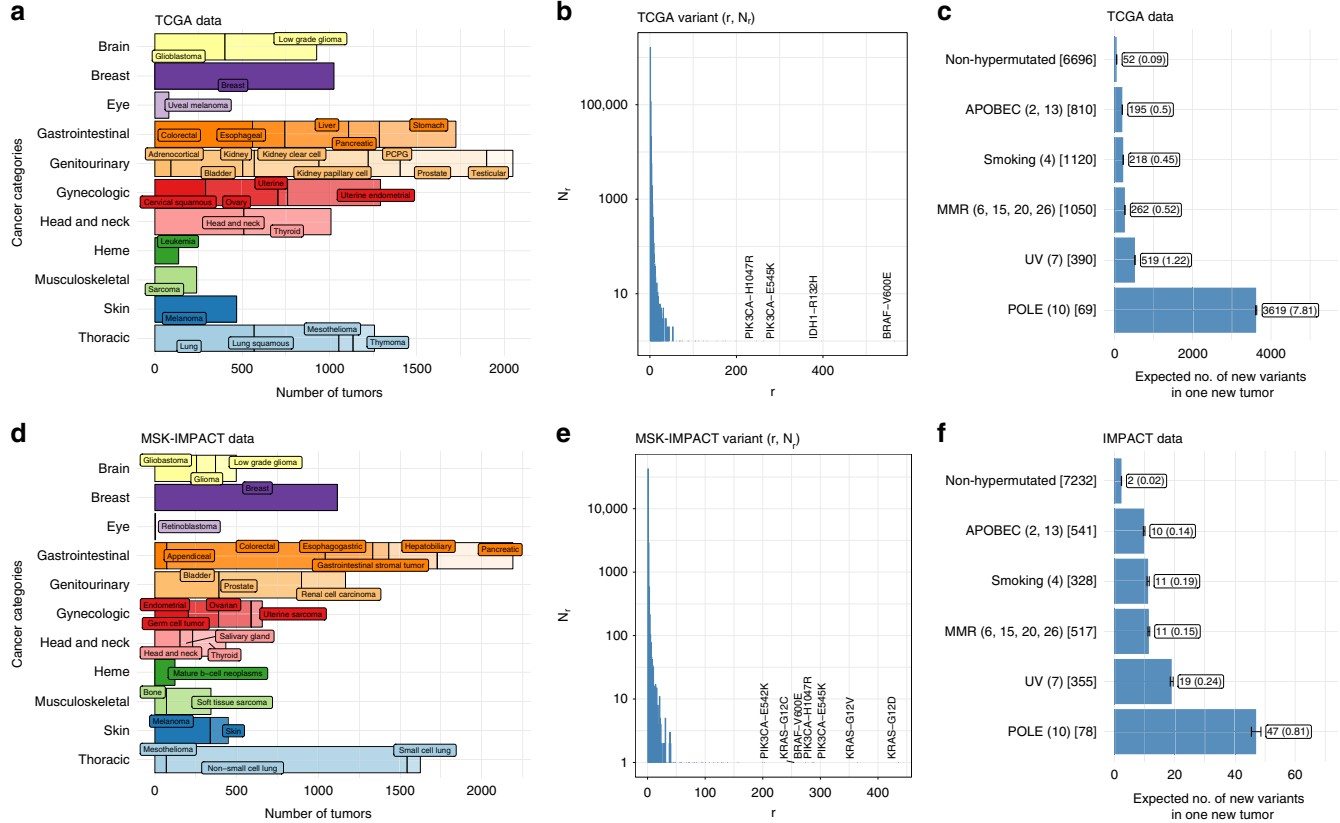

**Fig. 1** The TCGA and MSK-IMPACT data. **a**, **d** Tissue type compositions for TCGA data (**a**) and MSK-IMPACT data (**d**) show differences in the respective tissue-specific cohort sizes. **b**, **e** Variant frequencies (*r*) plotted against the associated number ($N_r$) of variants with that frequency in TCGA (**b**) and MSK-IMPACT (**e**) data. More than 90% of the variants in either dataset are singletons, i.e., appear only once. **c**, **f** The smoothed Good-Toulmin estimated average numbers of new variants (whole-exome variants for TCGA, and variants of the 410 paneled genes in MSK-IMPACT) in a new tumor together with its standard error obtained separately for each tumor subgroup in TCGA (**c**) and MSK-IMPACT (**f**) are displayed. The numbers inside the boxes are the respective estimates (outside parentheses) and their estimated standard errors (inside the parentheses), and the error bars correspond to ±2 standard error bounds. The numbers within parentheses in the vertical axes labels indicate the SBS numbers belonging to each dominant signature group (e.g., APOBEC corresponds to SBS numbers 2 and 3) and the numbers inside the square brackets denote the cohort sizes of the associated group in the TCGA and MSK-IMPACT data sets.

(*r* = 44 times) variants. By contrast *FAT1* has no hotspots while *PTEN* has an intermediate number.

Figure 3a displays our probability estimates by tumor site for a few selected common variants in *KRAS* and *PIK3CA*. These examples show that the probabilities of these variants can be highly tissue-type specific. Among all the observed variants in the 585 cancer genes annotated in OncoKB[16], a repository of curated cancer genes, *IDH1* R132H and *BRAF* V600E are the two most tissue-specific variants with NMIs of 0.18 and 0.13, respectively, followed by *GTF2I* L424H with an NMI of 0.08 (Supplementary Fig. 3). There are 15 variants with an NMI >0.03 (Supplementary Fig. 3). Figure 3a shows the common variants in *KRAS* with G12D and G12V being the two most lineage-dependent variants with highest probability of occurring in pancreatic cancer adenocarcinomas (PAAD), followed by colorectal cancer (COADREAD) and lung adenocarcinoma (LUAD). G12C is primarily associated with LUAD, whereas G12R occurs more exclusively in PAAD. In *PIK3CA*, the H1047R hot-spot has the highest probability of occurring in breast cancer (BRCA), whereas E545K has the highest occurrence in cervical cancer (CESC) and R88Q is more exclusive to endometrial cancer (UCEC).

Mapping these common variants in this way is relatively straightforward. The novelty in our strategy is to apply these ideas to previously unobserved variants. Figure 3b displays the corresponding probabilities of observing at least one previously

unobserved variant in a new tumor in selected genes by tumor site. [Note that in this and other figures oncogenes are in orange and tumor suppressor genes are in blue.] When encountering a new variant in *PTEN*, the three most likely tissue sources would be endometrial, uterine cancer, and glioblastoma. A total of 210 genes show substantial tissue specificity with an NMI greater than 0.01 (Fig. 3c). To put these numbers in perspective the orange histogram displays the null distribution of NMI values for the 585 cancer genes in OncoKB, assuming random assortment of variants to tissue types (see Supplementary Methods for details). This is contrasted with the observed histogram of NMI values in blue. Supplementary Figure 4 displays the genes that possess the largest degrees of tissue specificity. Our methods thus allow a systematic mapping of unseen variant probabilities that can be used toward understanding tissue type specificity and potentially for diagnosing the tissue of origin of cancers of unknown primary. It is of note that the tissue dependency of the common variants in *KRAS* and *PIK3CA* highlighted in Fig. 3a is widely recognized as clinically important, and so values of NMI > 0.02 in other genes are of credible clinical relevance also.

More generally, rare variant frequency profiles are observed to vary greatly across genes in the TCGA data. For cancer genes mutated in at least 3% of the TCGA cohort Fig. 4a displays the estimated probability of observing at least one new variant in a new tumor against the percentage singleton incidences. It is apparent

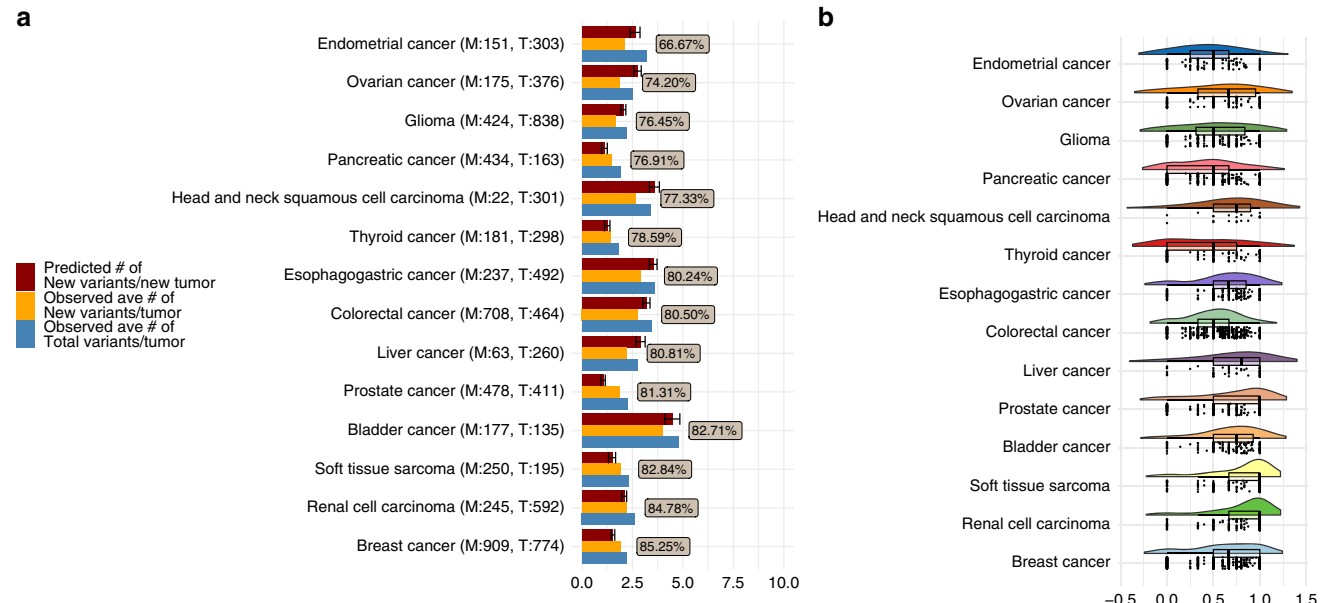

**Fig. 2** On average a substantial proportion (≥66%) of observed variants per individual in the MSK-IMPACT data are new (compared to the TCGA data). **a** the average observed number of variants (in blue), observed number of new variants (not in TCGA, in orange), and predicted (from the TCGA data, in dark-red) number of new variants with ±2 standard error bounds for the predictions (the error bars) per individual in the MSK-IMPACT cohort are plotted along the horizontal axis, separately for each tissue type (on the vertical axis). The percentages inside the boxes represent the ratios of the average observed number of variants to the average observed number of new variants per MSK-IMPACT tumor, and the numbers inside the parentheses in the vertical axis labels denote the respective cohort sizes (M: MSK-IMPACT, T: TCGA). **b** Distributions of the actual ratios of the number of new variants to the total number of variants per individual in the MSK-IMPACT cohort are plotted (along the horizontal axis) against tissue types (on the vertical axis) as grouped raincloud plots. The dots ("rain drops") represent the actual ratios, and the "rainclouds" and the box-plots summarize distributions of these ratios. All numbers shown in this figure correspond to tumors belonging to the non-hypermutated subgroups of the respective data sets.

from the figure that unseen variant probabilities are not in general related to the overall mutation frequency (characterized by sizes of the dots): a frequently mutated gene does not necessarily have a higher probability of producing a new or hitherto unseen variant, or vice versa. The probability of observing a new variant, however, is more strongly related to the proportion of the observed incidences caused by infrequent variants, in particular, percent singleton incidences. Figure 4a, b show, respectively, that the estimated probability of observing new variants and the predicted total number of new variants in a future cohort increase as a function of the percentage of singleton incidences in the gene. These figures also show that tumor suppressor genes (blue) harboring inactivating mutations tend to cluster toward the high end of singleton proportion scale (Fig. 4a, b). By contrast, for oncogenes (orange) with variants highly concentrated at hotspots, particularly *KRAS, BRAF, IDH1*, the probability of encountering new variants in future tumor samples tends to be low. While some genes are mutated primarily at a few hot-spots, other genes are mutated mostly at infrequent variants. As an illustration, the three genes shown in Supplementary Fig. 2 have similar mutation frequencies in the cohort (3–7%), but very different distributions of rare variants. For example, in *KRAS* only 5% (41 out of 477) of the variants have been observed once. In contrast, in *FAT1*, the preponderance (94%) of the variants are singletons while *PTEN* has an intermediate frequency. Consequently, our methods predict substantially fewer unseen variants in *KRAS* than in *FAT1* (see Fig. 4b), while the corresponding estimate of the probability of observing at least one new variant in a new tumor is noticeably higher for *FAT1* as compared to *KRAS* (Fig. 4a).

**Validation**. In order to validate our results, we used the TCGA data to obtain predictions of the gene-specific incidences of new variants in the MSK-IMPACT dataset. Specifically, for each gene

in the MSK-IMPACT panel, the Good-Turing estimated probability of observing at least one unseen variant (in a new tumor) based on TCGA data was compared with the observed relative frequency of tumors with at least one new variant in the MSK-IMPACT data (Fig. 4c), and the TCGA based predicted number of unseen variants was compared with the actual number of new variants observed in the MSK-IMPACT cohort (i.e. variants observed in MSK-IMPACT but not in TCGA, Fig. 4d). These figures show that the estimates are remarkably accurate, with Lin's concordance correlation coefficient reproducibility index being 0.93 and 0.92 respectively for the data in Fig. 4c, d respectively. This provides high confidence in the validity of our proposed variant probability estimation strategy. These analyses were repeated for each of the subgroups defined by the mutational signatures and the results are displayed in Supplementary Figs. 5 and 6. In general the predictions are highly accurate for APOBEC, MMR, UV, and smoking, with the exception being the POLE sub-group in which mutation frequency is over-predicted, though the sample sizes for this sub-group are small (69 in TCGA and 78 in MSK-IMPACT).

**Patterns of co-mutations across cancer types**. In order to study co-occurrence and mutual exclusivity of cancer genes, we applied the Good-Turing probability estimation strategy to gene pair frequencies. This analysis is restricted to the non-hypermutated tumors and is gene specific, not variant specific, as our data exploration shows that co-mutation analysis is feasible only at the gene level.

We ranked gene-pair co-mutations according their NMI with tissue type, and noticed that there are 390 gene-pairs (0.2%) with NMI >0.01. Figure 5a shows the top gene pairs are co-mutated in a highly lineage-dependent manner. Co-occurrences of *APC* with *KRAS and TP53* are notable in colorectal cancers as evidenced by

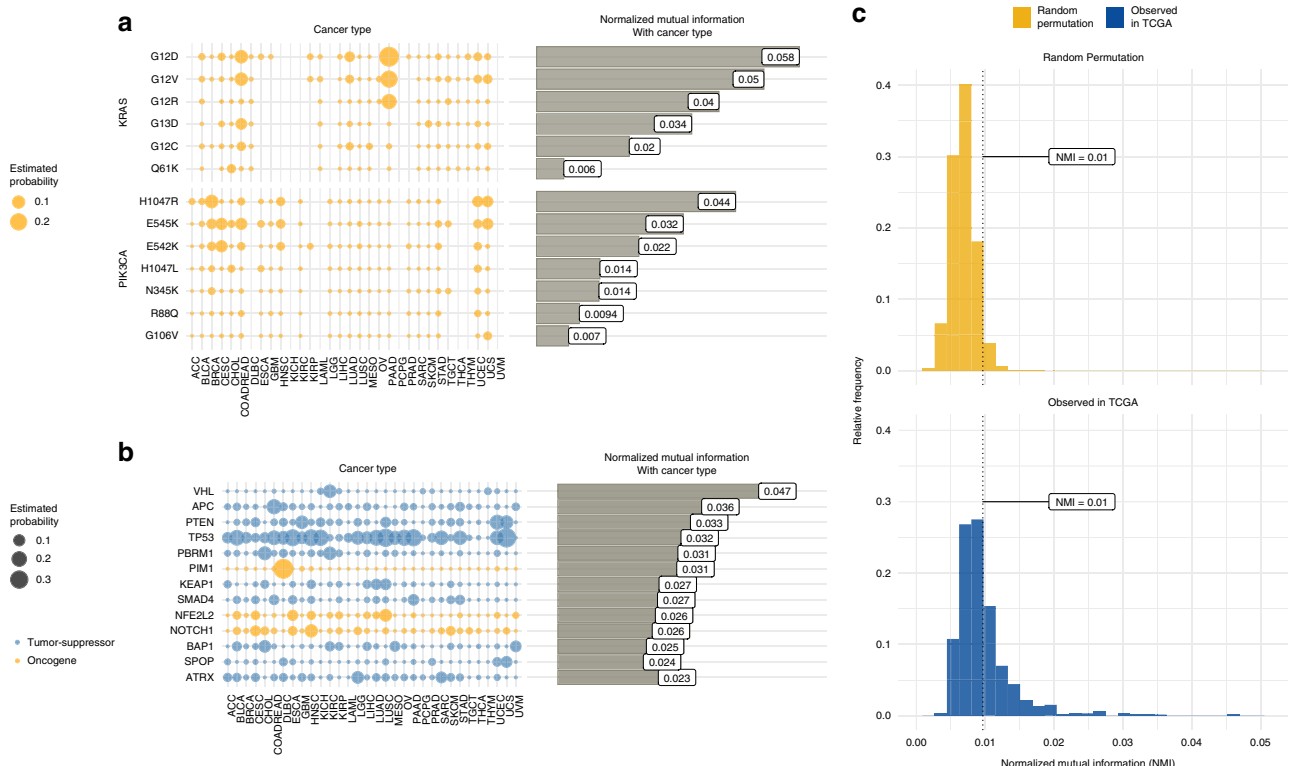

**Fig. 3** Estimating gene and tissue-type specific variant probabilities. **a**, **b** Gene and tissue-type specific probabilities of encountering a previously seen variant (**a**) and at least one hitherto unseen variant (**b**) in a new tumor differ substantially across genes and tissue-types. The dependencies between the tissue type and the occurrences of gene specific variants, both previously observed (**a**), and hitherto unobserved (**b**), as quantified by Mutual Information, vary substantially across genes. Oncogenes are shown in orange, tumor-suppressor genes are in blue, and non-annotated (in OncoKB list) genes are shown in gray. **c** The tissue specificity for observing at least one new variant in a gene in TCGA data, as quantified by NMI values (blue histogram), shows a distinct right shift as compared to the null reference NMI values obtained from random variant-tissue allocations (orange histogram). The dotted vertical line corresponds to the 95th percentile of the null reference NMIs.

the high frequencies (large circles) and consistent with the role of APC in early development of colorectal cancer[17,18]. We then searched for lineage-dependent co-mutation pairs for each individual cancer type. Here our NMI measure was adapted to identify the extent to which a co-mutation is elevated in an index cancer type compared with the average of all other cancer types (see Supplementary Methods). Supplementary Figure 7a shows the top five gene pairs having the largest NMI values associated with each cancer type, further highlighting the co-occurrences of IDH1 with TP53, ATRX, and CIC in low grade glioma, and co-occurrences of CTNNB1 with PTEN and PIK3CA in endometrial cancer. Similar analyses were performed to search for lineage-dependent mutually exclusive gene pairs. We computed Good-Turing probability estimates for these pairs based on their mutual exclusive frequencies, and top gene pairs with high NMIs are shown in Fig. 5b, confirming some of the known clinically relevant mutual exclusive patterns including RAS and RAF mutations[19–21]. Supplementary Figure 7b displays the top mutually exclusive gene pairs having the largest mutual information associated with each cancer type, further highlighting IDH1/2 mutations in glioma, GNA11 and GNAQ in uveal melanoma. Overall, these results suggest that co-occurring and mutually exclusive gene pairings may further define our ability to predict tissue type.

## Discussion
Our results shed light on the potential information content in the vast trove of mutations that occur at genetic loci with very low

probability of occurrence. Fully 92% of the 1,788,153 distinct non-synonymous variants that have been observed in TCGA have only been observed once and most new tumors harbor mutations that have never been observed before, even when using restricted sequencing panels. Are the preponderance of these mutational events irrelevant consequences of genetic instability, or do they contain important signals that could be harnessed for clinically relevant purposes? While we have not addressed this question definitively, our results do suggest that there could be considerable information content in this submerged portion of this genomic iceberg. We have shown that when we use established statistical methods to estimate the frequencies of unseen variants and their related probabilities of occurrence and mutual exclusivity on a gene-specific level we observe substantial variation among genes and among the anatomical sites from which the tumors emerged, suggesting that these variants have the potential to be clinically informative.

How could this information be used in a clinical context? The results are potentially valuable for the task of identifying the primary site of tumors of unknown origin, or when evidence of a tumor is identified in, say, circulating tumor DNA. In this context the congruence of the pattern of mutations observed with different tumor types provides the crucial information, but this depends on knowledge of the mutation probabilities expected in the candidate tumor types. Since the preponderance of somatic mutations observed in any given tumor are either rare or previously unobserved the methods presented in this article are essential for investigating this tissue-type congruence. Another context in which the results are useful is in the task of testing

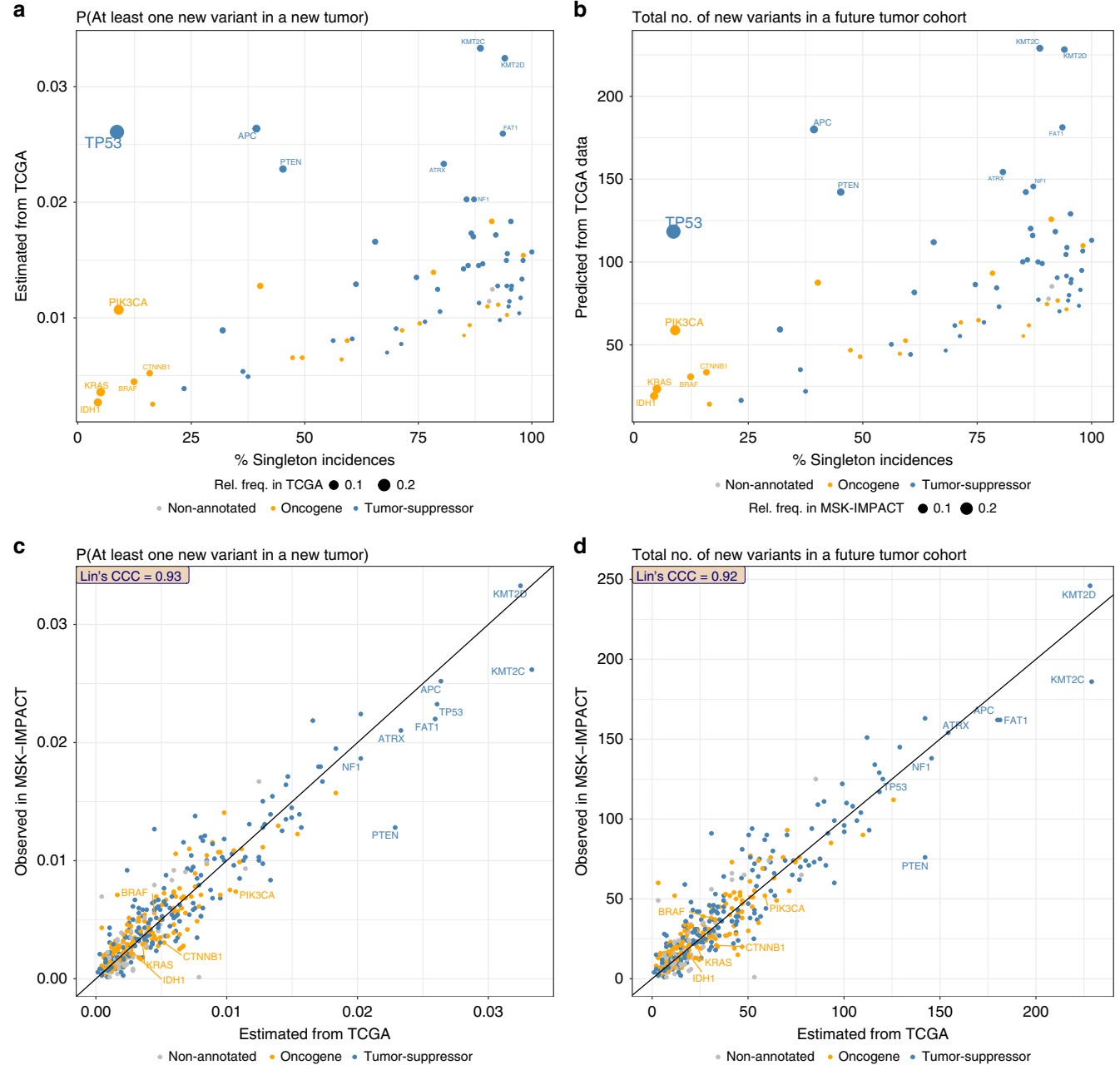

**Fig. 4** The estimated numbers and probabilities associated with unseen variants are remarkably accurate. **a** TCGA data based estimated probability of observing at least one new variant in a new tumor, plotted against the percent singleton incidences (in TCGA data) for each *frequent* (mutation rate in TCGA data ≥0.03) gene that is also in the MSK-IMPACT panel. **b** TCGA data based predicted number of new variants in a future sequencing cohort of size 7254 (the number of non-hypermutated tumors in MSK-IMPACT cohort), plotted against the percent singleton incidences (in TCGA data) for each *frequent* gene in MSK-IMPACT panel. **c** The TCGA estimated probability (the quantity along the vertical axis of panel **a**), plotted against the observed relative frequency of at least one new variant in a new tumor in MSK-IMPACT data for *all* MSK-IMPACT genes. **d** The TCGA predicted number of new variants in a future study with 7,254 tumors (the quantity along the vertical axis of panel **b**), plotted against the observed number of new variants in MSK-IMPACT data, for *all* MSK-IMPACT genes. Oncogenes are shown in orange, tumor-suppressor genes are in blue, and non-annotated (in OncoKB list) genes are shown in gray. All estimates in this figure correspond to tumors in the non-hypermutated subgroups of the respective datasets.

pairs of tumors in the same patient for clonal relatedness, i.e. testing whether one tumor is a metastasis of the other or whether the tumors arose independently, a diagnosis that can have clinical implications[22]. The key information supporting a diagnosis of clonal relatedness is the presence of identical mutations in the two tumors, but the significance of such an observation depends strongly on how common or rare is the mutational event[23]. Furthermore, it is common in clonality studies to encounter a new variant that has not been previously seen, yet shared by a

pair of tumors, that could provide essential evidence for clonality. Therefore, a strategy for estimating the new variant probability is essential.

Our analyses have notable limitations. First, the product binomial model considered for the individual variant probabilities assumes that different variants occur independently. This assumption is simplistic and is known to be violated in practice, as some variants can be preferentially co-mutated and/or mutually exclusive[24]. However, as our validation experiment in

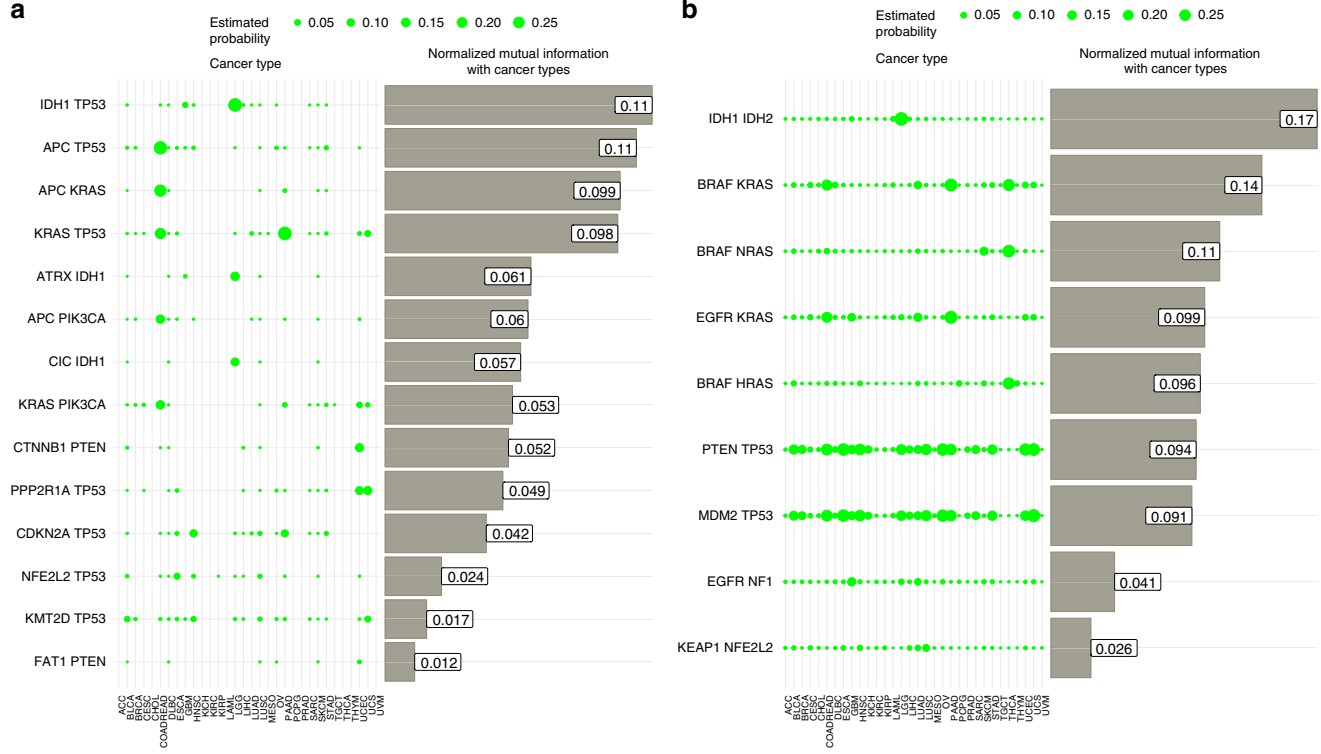

**Fig. 5** The Good-Turing estimation of lineage-dependent gene co-mutation and mutual exclusivity probabilities. The probability of observing a gene co-mutation (**a**) or a mutation in one or the other but not both (**b**), as well as the dependency between the tissue type and these occurrence probabilities, as quantified by Normalized Mutual Information, varies substantially across gene pairs.

the MSK-IMPACT data demonstrates (see Fig. 4c, d), the model provides remarkably accurate results regardless of this limitation. We also showed that accurate predictions are possible when tumors are classified into distinctive tumor categories defined by hypermutation, except for the ultra-hypermutated POLE signature. Given the highly specific mutational signature of POLE, a different modeling approach is needed as future work. Also, we were able to easily adapt the methods to analyze co-mutations in gene pairs, and these data also show interesting and informative discrimination by tissue type. Second, our analysis focuses on non-synonymous single-nucleotide variants. Other types of alterations such as small insertions/deletions, and gene fusions were not considered. These alterations only account for a small fraction of the total pool, and thus are not likely to impact our conclusions significantly. Third, the validation studies are based solely on the 410 genes in the MSK-IMPACT cohort. These are primarily cancer genes and cannot be considered representative of the complete genome.

In summary, we believe that there is strong potential for using statistical methods to harness information content in the vast preponderance of mutations that occur at "rare" mutational loci. We have shown how to estimate the numbers of unseen variants and their corresponding probabilities of occurrence and have identified genes where these probabilities vary substantially by tumor site, offering evidence of their potential for classifying cancers of unknown primary or tumors detected in ctDNA. We believe that more intense investigation of the properties of this "submerged" portion of the iceberg has potential to yield consequential information of clinical relevance for cancer.

## Data availability

The two data sets used in this study are publicly available. The TCGA data set is available at https://portal.gdc.cancer.gov/, and the MSK-IMPACT data set is available at http://cbioportal.org/msk-impact. An R package containing these data sets and software

implementation of the methods used in this study has been released in the public domain (https://github.com/c7rishi/variantprobs). There is no restriction to the availability of the data used in this study.

## Code availability

A software implementation of the methods used in this study has been released in the public domain in the form of an R package entitled variantprobs[25]. The package contains functions for performing Good-Turing probability estimation and smooth Good-Toulmin expected number of unseen variants $\Delta(t)$ estimation for any $t > 0$. The exact datasets used in this study, obtained by filtering (i.e., keeping only the non-synonymous single nucleotide variants) the TCGA data and the MSK-IMPACT data, are also stored in the R package.

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

## Acknowledgements

The research was supported by the National Cancer Institute, Award CA008748. The authors express their gratitude towards the reviewers whose insightful reviews and suggestions helped improve the article.

## Author contributions

S.C., C.B.B. and R.S. designed the research. S.C. made software implementations and analyzed the data. A.A. performed the mutation signature analyses. S.C., C.B.B. and R.S. wrote the paper.

## Competing interests

The authors declare no competing interests.
