## [Peer Review File · Nature Communications]

Reviewers' comments:

Reviewer #1 (Remarks to the Author):

Chakraborty et al. applied the Good-Turing estimator to estimate the number of unseen variants in tumor genomes for all possible future samples and for a specific future sample. They found that the probability of encountering a previous seen (or unseen) variant is tissue-type specific, which would be utilized for identifying the tissue of origin for circulating tumor DNA or metastasis. Furthermore, they showed that pattern of variants can distinguish cancer driver genes from passenger genes. The study has several interesting and potentially important findings. The paper can be improved in several aspects.

A. It was estimated that about 11.37 million (M) non-synonymous mutations would be eventually identified. Given roughly 26.9M possible non-synonymous (based on RefCDS) sites in human genome, an interesting question arises why more than half of possible non-synonymous sites exempt from be detected of a somatic mutation, even with infinite number of tumor samples? Alternatively, I hypothesized that this predicted number (11.37M) may be underestimated: TCGA samples were filtered out somatic mutations with $VCF < 0.1$, most of which were likely subclonal and singletons; and if more subclonal singletons are identified, the predicted number of non-synonymous mutations being eventually identified will increase (by Chao Estimator). It's of interest to conduct simulation studies to evaluate the impact of VCF cutoff on the number of singletons and consequently on the number of non-synonymous mutations being eventually identified. In addition, it's of interest to add more discussion if this analysis would answer the question "will each possible non-synonymous site be mutated in some tumors?"

B. Stratification analysis. The current estimations of somatic variant richness lump all somatic variants together. However, it's well known that the background mutation rate of somatic variants varies significantly across context and genomic regions. It's of interest to conduct analysis for mutations due to specific mutational signatures: COSMIC SBS1 (deamination of 5-methylcytosine), SBS4 (smoking) and SBS2/13 (APOBEC);

C. Mutual information (MI). The MI is well defined but it's value of clinical application or risk-stratification is hard to evaluate. Most of MI values in the paper are less than 0.1; are they clinically significant?

D. Oncogene vs tumor suppressor gene (TSG). It's well recognized that oncogenes in general harbor mutation hot-spots while TSG not. It's of interest to comment results in Figure 3 and Figure 4 w.r.t oncogene vs TSG.

E. Would it possible to evaluate patterns of mutually exclusive across cancer types?

F. Results on figure 2a. Is it possible to add the predicted “ave no of new variants” of MSK-IMPACT based on TCGA with confidence intervals and compare the predicted ones with the observed ones?

G. Results on figure 6. Would the ROC curves on 6d compares to ROC curves based on 1) dN/dS ratio (obtained from dndscv) and 2) measurements of clustering pattern (e.g. Vogelstein’s 20/20 rule; or clustering score of OncodriveCLUST)? Would combination of patterns of variants with dN/dS ratio and/or clustering pattern further improve AUC?

H. Minor comments: 1) Would results be changed if hypermutated samples (~8% in TCGA) were removed? 2) How the 81 passenger genes in figure 6 were selected? 3) Results on figure 4. Would the diagonal lines be added to the figures? It seems that estimated probabilities from TCGA are smaller than ones observed in MSK-IMPACT.

Reviewer #2 (Remarks to the Author):

The authors present an interesting analysis of the chances of seeing new variants as tumor cohorts increase in size. The analysis is based on the Good-Turing estimator of the probability of a new variant, which is then applied to the TCGA and MSK-Impact data sets. There are some interesting findings, and the analysis could be of interest to the readership of Nature Communications, although it needs some refining as detailed below -

1. In a sense, we already know the number of unobserved non-synonymous substitutions - the exome is finite and known. It is ~30Mb, and each nucleotide within the exome can be substituted by 3 other bases. That implies that there are overall ~90 million coding substitutions possible, which allowing for redundancy in the genetic code means maybe 65-70 million non-synonymous subs. This is considerably higher than the authors estimate (11.37 million). So, why are the authors so far off the true value? My belief is that it comes down to points 2 and 3 below, and overly simplifying assumptions in the estimator that do not capture the complexity of the actual data.

2. Failure to distinguish passenger and driver mutations adequately - Clearly some of the high recurrence variants are drivers; many of the singletons are passengers. The genes in the MSK-Impact gene-set are enriched for cancer genes; those in the TCGA exomes are enriched for non-cancer genes. I do not know whether the Good-Turing estimator etc allow for different classes of objects, following radically different distributions of recurrence, but this would seem an important consideration.

3. Failure to account for differences in mutational signatures - Numerically, many of the variants in the TCGA exomes will come from a small number of hypermutators (particularly POLE mutated cancers). POLE tumors in particular have a remarkably spiky, specific mutational signature that will make for a very different profile of recurrence versus singletons than hypermutators with flatter mutational profiles (BRCA1/2 for example).

I am not sure what the effects of these biases would be on the authors' calculations, but it would seem as a minimum that they should use simulations to test this - it is relatively straightforward to simulate mutations from mutational signatures / distributions of passenger and driver mutations - this could be done with a variety of simulated scenarios to assess the effects of these factors on their conclusions.

Review signed by Peter Campbell

Concerns of Both Reviewers

The initial issue raised by both reviewers concerns our estimate of the total number of previously unseen non-synonymous variants. We estimated a lower bound (the Chao method) of this to be in the region of 11.37 million, and both reviewers point out that the true number is likely multiples of this total. On reflection we recognize that it was a distraction to emphasize this estimate since it has only peripheral relevance to the thesis of our article. Our thesis is that the rates of occurrence of previously unseen variants at the gene level can be highly tissue specific, and that these rates are strongly predictable using various adaptations of the “Good-Turing” methodology. One only needs an estimate of the total number of unseen variants (in a specific gene) to estimate the probabilities of occurrence of **each individual variant**. However, these entities, and the estimate of the numbers of unseen variants on which they are based, are not easily validated. What we can estimate, and what we did validate, is the probability of observing **any** previously unseen variant in a specific gene, for an individual tumor or for a validation sample of a defined (limited) size. This, in essence, represents the collective probability related to all of the unseen variants and it can be estimated using an asymptotic approximation that involves solely the number of singletons observed in the training sample but does not involve the number of unseen variants (N_0) (see equation 5 in Supplementary Methods). We added this discussion in “Methods Overview”. The validity of this probability estimate (derived from TCGA data) can be evaluated by comparing it with the relative frequency of cases harboring a new variant in each gene

(using the MSK-IMPACT validation data), and this is what was (and still is) presented in Figure 4c, a key figure in the article, showing a concordance correlation of 0.93 between the predicted unseen variant probability and observed frequency in the validation cohort. The combination of strong tissue dependency of these probabilities (Figure 3b and c) and the exceptional validity is our key finding, in that it demonstrates the potential actionability of the information content of these rare/unseen variants for tissue diagnostic purposes. Note that we also estimated and validated a strongly related quantity, the total number of previously unseen variants predicted to occur in the validation dataset (given its sample size), and the strong validation of this quantity is shown in Figure 4d. In short, in our prior submission we gave inappropriate attention to the estimation of the total number of unseen variants. In the new version we have eliminated all material related to estimation of the total number of unseen variants.

The second issue raised by both reviewers was the speculation that the probability estimates would vary substantially on the basis of whether or not the tumors possess known mutational signatures related to hypermutation. The reviewers are absolutely correct in this regard. In the revision we have performed mutational signature analysis¹ and categorized each sample into one of six categories: non-hypermutated, APOBEC(SBS 2/13), Smoking-associated (SBS 4), MMR (SBS 6,15,20,26), UV (SBS 7) and POLE (SBS 10). The five SBS signatures are assigned according to the Sanger COSMIC mutational signature annotation. The non-hypermutated groups in TCGA and MSK-IMPACT were defined as tumors not in one of these 5 signature categories and with a total mutation burden less than 500 and 38 respectively. The hypermutation threshold for TCGA tumors was obtained from Martincorena, et al.², and that for MSK-IMPACT tumor was obtained by multiplying the TCGA threshold with the ratio of median total mutation burden in MSK-IMPACT (4) to that in TCGA (54). We then used our methodology to predict, based on TCGA data, the proportion of tumors in the validation MSK-IMPACT dataset with at least one previously unseen variant in a specific gene and contrasted these with the corresponding observed relative frequencies (by gene) for each of these 6 strata separately. The results, in Supplementary Figure 4, show impressive predictive accuracy, except for the (small, only 0.7% of cases) POLE group where our formula seems to somewhat over-predict this quantity, and for the smoking subgroup where there is modest under-prediction. Of course the preponderance of tumors (66%) are in the non-hypermutated group and in the main body of the article and the main tables we now focus attention on this non-hypermutated subset of tumors.

In the following we respond to the individual reviewer comments. Reviewers' original comments are pasted in **bold fonts**, our responses are in normal fonts. Changes in the main text are in **red**.

Reviewer #1 (Remarks to the Author):

A. It was estimated that about 11.37 million (M) non-synonymous mutations would be eventually identified. Given roughly 26.9M possible non-synonymous (based on RefCDS) sites in human genome, an interesting question arises why more than half of possible non-synonymous sites exempt from be detected of a somatic mutation, even with infinite number of tumor samples? Alternatively, I

hypothesized that this predicted number (11.37M) may be underestimated: TCGA samples were filtered out somatic mutations with $VCF < 0.1$, most of which were likely subclonal and singletons; and if more subclonal singletons are identified, the predicted number of non-synonymous mutations being eventually identified will increase (by Chao Estimator). It's of interest to conduct simulation studies to evaluate the impact of VCF cutoff on the number of singletons and consequently on the number of non-synonymous mutations being eventually identified. In addition, it's of interest to add more discussion if this analysis would answer the question "will each possible non-synonymous site be mutated in some tumors?"

Response: See first issue in response to "Concerns of Both Reviewers" above.

B. Stratification analysis. The current estimations of somatic variant richness lump all somatic variants together. However, it's well known that the background mutation rate of somatic variants varies significantly across context and genomic regions. It's of interest to conducted analysis for mutations due to specific mutational signatures: COSMIC SBS1(deamination of 5-methylcytosine), SBS4(smoking) and SBS2/13(APOBEC);

Response: See second issue in response to "Concerns of Both Reviewers" above.

C. Mutual information (MI). The MI is well defined but it's value of clinical application or risk-stratification is hard to evaluate. Most of MI values in the paper are less than 0.1; are they clinically significant?

Response: We modified our approach to use a normalized mutual information (NMI) in the revised version for easier interpretation and comparison across features. Although the NMI is related to the correlation coefficient in the case of bivariate normal data, it still does not have an intuitive interpretation for practical utility in discrete data like ours (variant-tissue type association). However, we can obtain some benchmark numbers to provide a sense of how large NMI must be to have clinical value. Figure 3a presents the well-known tissue-specific variants G12D, G12V, G13D and G12R in *KRAS* and H1047R E545K and E542K in *PIK3CA*. These are widely recognized as clinically important variants that occur in a highly lineage-dependent manner. They have NMIs in the range of 0.02 to 0.06. We point out that an NMI in this range can be considered of potential biological and clinical relevance (end of 3rd paragraph in this section). In addition, the significant shift in the NMI distribution to the right of the null distribution as shown in Figure 3c suggests that there is a great amount of variant-tissue association that can be extracted and utilized. We added a conceptual description of the normalized mutual information in the "Methods Overview" section with a more detailed technical description in Supplementary Methods, and we included the preceding discussion on biological and clinical relevance to the Results section on page 10.

D. Oncogene vs tumor suppressor gene (TSG). It's well recognized that oncogenes in general harbor mutation hot-spots while TSG not. It's of interest to comment results in Figure 3 and Figure 4 w.r.t oncogene vs TSG.

Response: We have now distinguished oncogene and tumor suppressor genes using different colors in revised Figure 3 and 4. We adopted the oncogene and TSG labeling using the OncoKB annotation³. This leads to the observation that tumor suppressor genes (blue) harboring inactivating mutations tend to cluster toward the high end of singleton proportion scale (%Singleton on x-axis) (Figure 4 a and b). By contrast, for oncogenes (yellow) with variants highly concentrated at hotspots, particularly *KRAS*, *BRAF*, *IDH1*, the probability of encountering new variants in future tumor samples tends to be low. This discussion is added to the revised manuscript in the Results section on page 10/11.

E. Would it possible to evaluate patterns of mutually exclusive across cancer types?

Response: We have now included a panel for tissue-dependent mutually exclusive gene pairs in Figure 5b along with discussion of the results on page 12.

F. Results on figure 2a. Is it possible to add the predicted “ave no of new variants” of MSK-IMPACT based on TCGA with confidence intervals and compare the predicted ones with the observed ones?

Response: Added. Please see revised Figure 2a. This shows the predictions are generally quite good for non-hypermutated tumors. We left out melanoma and lung cancers from the revised figure, as these two cancer categories are primarily associated with tumors possessing UV and Smoking signatures respectively, and the non-hypermutated group used in this analysis excluded such tumors.

G. Results on figure 6. Would the ROC curves on 6d compares to ROC curves based on 1) dN/dS ratio (obtained from dndscv) and 2) measurements of clustering pattern (e.g. Vogelstein’s 20/20 rule; or clustering score of OncodriveCLUST)? Would combination of patterns of variants with dN/dS ratio and/or clustering pattern further improve AUC?

Response: We took the reviewers suggestion and applied the dN/dS ratio method by Martincorena et al.² to distinguish driver vs passenger genes presented in the original Figure 6. We used linear discriminant analysis (LDA) to combine the missense, nonsense, splice site, and indel dN/dS ratio estimates into a single linear score. We also included the proportion of recurrent missense mutations and proportion of inactivating mutations as defined in the 20/20 rule by Vogelstein et al⁴. As shown in the Figure below, both achieved high discriminant power, with an 82% AUC. By comparison, the proportion of singleton and maxr are less discriminating. Adding these to dN/dS or the 20/20 rules did not lead to any notable increase in the AUC. Given these additional analyses, we acknowledge that although the frequency vector (r, N_r) can accurately predict the number of unseen variants, they are not competitively powerful predictors for distinguishing driver vs passenger genes, and do not add additional value to the existing methods. On this basis we elected to remove Figure 6 and the related discussion of this issue from the manuscript.

H. Minor comments:

1) Would results be changed if hypermutated samples (~8% in TCGA) were removed?

Response: As noted above, in the new version we have elected to focus attention on non-hypermutated tumors. Figure 4d in the revised version now shows that the predicted and the actual observed number of new variants in the validation cohort are highly concordant, with a concordance correlation of 0.92.

2) How the 81 passenger genes in figure 6 were selected?

Response: Passenger genes in Figure 6 were selected based on a combined score including DNA replication timing, gene expression, and chromatin structure using data published in Lawrence et al⁵. Top ranking genes with high background mutation rates as predicted by these variables were chosen as passenger genes. We note that Figure 6 has been removed from the revised version for reasons noted above.

3) Results on figure 4. Would the diagonal lines be added to the figures? It seems that estimated probabilities from TCGA are smaller than ones observed in MSK-IMPACT.

Response: We have added diagonal lines and concordance correlation on each graph.

Reviewer #2 (Remarks to the Author):

The authors present an interesting analysis of the chances of seeing new variants as tumor cohorts increase in size. The analysis is based on the Good-Turing estimator of the probability of a new variant, which is then applied to the TCGA and MSK-Impact data sets. There are some interesting findings, and the analysis could be of interest to the readership of Nature Communications, although it needs some refining as detailed below –

1. In a sense, we already know the number of unobserved non-synonymous substitutions - the exome is finite and known. It is ~30Mb, and each nucleotide within the exome can be substituted by 3 other bases. That implies that there are overall ~90 million coding substitutions possible, which allowing for

redundancy in the genetic code means maybe 65-70 million non-synonymous subs. This is considerably higher than the authors estimate (11.37 million). So, why are the authors so far off the true value? My belief is that it comes down to points 2 and 3 below, and overly simplifying assumptions in the estimator that do not capture the complexity of the actual data.

Response: Please see our response to this comment in the first paragraph of “Response to Both Reviewers” above.

2. Failure to distinguish passenger and driver mutations adequately - Clearly some of the high recurrence variants are drivers; many of the singletons are passengers. The genes in the MSK-Impact gene-set are enriched for cancer genes; those in the TCGA exomes are enriched for non-cancer genes. I do not know whether the Good-Turing estimator etc allow for different classes of objects, following radically different distributions of recurrence, but this would seem an important consideration.

Response: The reviewer raises two issues here. The first is the fact that the genes in MSK-IMPACT are not representative of all genes, being “enriched” for cancer genes. This is undoubtedly true. We have used MSK-IMPACT as a validation dataset due to its availability and convenience, and we feel that the validation of our predictions are very impressive (Figure 4c). However, we now acknowledge more explicitly that the accuracy of our probability estimates in the much larger constellation of genes in the exome remains to be validated (3rd paragraph of Discussion). The second issue raised is whether we can refine our analysis to study “different classes of objects”. In the response to point 3 below, described earlier in the second paragraph of “Response to Both Reviewers” we describe our sub-set analyses of tumors defined by mutational signatures. However, we have also clarified the distinction in the results between oncogenes and tumor suppressor genes, using color codes in Figures 3a-b and 4a-d. The results show quite distinctive patterns, notably the typically lower % frequency of singleton variants among oncogenes and the correspondingly lower probabilities of encountering previously unseen variants in a new tumor.

3. Failure to account for differences in mutational signatures - Numerically, many of the variants in the TCGA exomes will come from a small number of hypermutators (particularly POLE mutated cancers). POLE tumors in particular have a remarkably spiky, specific mutational signature that will make for a very different profile of recurrence versus singletons than hypermutators with flatter mutational profiles (BRCA1/2 for example).

I am not sure what the effects of these biases would be on the authors' calculations, but it would seem as a minimum that they should use simulations to test this - it is relatively straightforward to simulate mutations from mutational signatures / distributions of passenger and driver mutations - this could be done with a variety of simulated scenarios to assess the effects of these factors on their conclusions.

Response: Please see the second paragraph in our response to “Concerns of Both Reviewers” above for our response to this concern.

Thank you for your consideration and we look forward to hearing from you.

References

1. Alexandrov, L. B. *et al.* Signatures of mutational processes in human cancer. *Nature* (2013). doi:10.1038/nature12477
2. Martincorena, I. *et al.* Universal Patterns of Selection in Cancer and Somatic Tissues. *Cell* (2017). doi:10.1016/j.cell.2017.09.042
3. Chakravarty, D. *et al.* OncoKB: a precision oncology knowledge base. *JCO Precis. Oncol.* **1**, 1–16 (2017).
4. Vogelstein, B. *et al.* Cancer genome landscapes. *Science* (2013). doi:10.1126/science.1235122
5. Lawrence, M. S. *et al.* Mutational heterogeneity in cancer and the search for new cancer-associated genes. *Nature* **499**, 214 (2013).

REVIEWERS' COMMENTS:

Reviewer #1 (Remarks to the Author):

Chakraborty et al. have addressed review comments well. A number of minor comments are listed below to improve the presentation of paper:

- A. Is mutational signature analysis carried out de novo or by pre-specifying mutational signatures in advance?
- B. Are the mutation burdens (in unit of mutations/MB) comparable between MSK-IMPACT and TCGA for each cancer type?
- C. Line 136: is negative sign missing in the formula, compared to eq (5) in supp. methods?
- D. Line 249: it seems that R88Q is not shown in fig3a.
- E. Line 230: could it be more specific about what are the probabilities estimated in this section?
- F. Fig1c,f: what are these numbers in parentheses and brackets? For the label of x-axis, should it be in one new tumor cohort study instead of "in one new tumor"?
- G. Fig 2a: what are these numbers in parentheses? Line 411: exchange the word position of "orange" and "blue"? Why the predicted numbers are much lower for some cancer types, e.g. germ cell tumor and thyroid cancer?

Review signed by Bin Zhu

Reviewer #2 (Remarks to the Author):

The authors have responded well to my comments, and I have no further suggestions.

Responses to reviewers' comments

We sincerely thank the reviewers for their insightful reviews and helpful suggestions that have helped us substantially improve our manuscript, both during the previous revision and the current. We have taken all of their comments into account while revising our manuscript. In the following we provide point-by-point response to each comment made by the reviewers. The reviewers' original comments are pasted in black fonts, our responses are in blue, and changes in the main text are in red.

REVIEWERS' COMMENTS:

Reviewer #1 (Remarks to the Author):

Chakraborty et al. have addressed review comments well. A number of minor comments are listed below to improve the presentation of paper:

- A. Is mutational signature analysis carried out de novo or by pre-specifying mutational signatures in advance?

Response: The mutation signature analysis is carried out by pre-specifying mutational signatures in advance, using the algorithm outlined in Zehir et al.¹ and the implementation available at <https://github.com/mskcc/mutation-signatures>. We have added these details in the Supplementary Material.

- B. Are the mutation burdens (in unit of mutations/MB) comparable between MSK-IMPACT and TCGA for each cancer type?

Response: We generated boxplots of the total mutation burden per tumor type in the TCGA cohort and the MSK-IMPACT cohort and displayed them side-by-side in Supplementary Figure 1. The comparison is shown for all tumors, restricting to the 410 MSK-IMPACT genes. The figure shows that the total mutation burdens are largely comparable for the MSK-IMPACT and the TCGA tumors. We have made a comment on this comparison on page 8 of the revised manuscript.

- C. Line 136: is negative sign missing in the formula, compared to eq (5) in supp. methods?

Response: Indeed, the negative sign was missing. We thank the reviewer for pointing out the typo; it has been corrected in the revision (p. 5).

- D. Line 249: it seems that R88Q is not shown in fig3a.

Response: We have added R88Q to the updated figure.

- E. Line 230: could it be more specific about what are the probabilities estimated in this section?

Response: We have specified that our interest in this section lies in probability estimation of individual mutations occurring at a gene, with major emphasis on rare and hitherto unobserved mutations (p. 9).

- F. Fig1c,f: what are these numbers in parentheses and brackets? For the label of x-axis, should it be in one new tumor cohort study instead of “in one new tumor”?

Response: We have updated the legend with explanations for these numbers (p. 18). The numbers within parentheses in the vertical axes labels indicate the SBS numbers belonging to each dominant signature group as annotated in COSMIC at <https://cancer.sanger.ac.uk/cosmic/signatures/SBS/> (e.g., APOBEC corresponds to signature numbers 2 and 3) and the numbers inside the square brackets denote the cohort sizes of the associated group.

The x-axis is indeed the expected number of variants in *one new tumor* (belonging to the associated group), which aids comparison across different tumor groups.

- G. Fig 2a: what are these numbers in parentheses? Line 411: exchange the word position of “orange” and “blue”? Why the predicted numbers are much lower for some cancer types, e.g. germ cell tumor and thyroid cancer?

Response: The numbers in parentheses are the sample size in MSK-IMPACT (M) and in TCGA (T). The legend has been updated with these details, and we have also exchanged the word positions of orange and blue (p. 18).

We thank the reviewer for pointing out the low predicted number of new mutations, particularly in germ cell tumor and in thyroid cancer. Upon careful reviewing, we realized there was a tumor-type matching error on thyroid cancer between the two data sets. This has been corrected and the updated results are shown in revised Figure 2. In germ cell cancer, we found the match was not one-to-one after obtaining more detailed cancer type annotation from the MSK-IMPACT dataset. In this case, the TCGA cohort consists of primarily testicular germ cell cancer whereas the MSK-IMPACT cohort contains a mixture of diverse types of germ cell tumors. Therefore we have removed germ cell tumor from revised Figure 2. Note that Figure 2 contains only the most frequent tumor types and this is now clarified in the manuscript (p. 9). We went through each tumor type category and further refined the head and neck squamous cell cancer and the liver cancer (hepatocellular carcinoma) category to ensure a close match. The rest remain the same. We thank the reviewer for bringing this to our attention.

Review signed by Bin Zhu

Reviewer #2 (Remarks to the Author):

The authors have responded well to my comments, and I have no further suggestions.

References

1. Zehir, A. *et al.* Mutational landscape of metastatic cancer revealed from prospective clinical sequencing of 10,000 patients. *Nat. Med.* (2017). doi:10.1038/nm.4333